# Examining Gender Differences and Their Associations Among Psychosocial Distress, Social Support, and Financial Well-Being of Informal Caregivers of Older Adults in the Rural Northcentral United States

**DOI:** 10.3390/healthcare14010017

**Published:** 2025-12-20

**Authors:** Nasreen Lalani, Evans Appiah Osei, Zihan Xu

**Affiliations:** 1School of Nursing, Purdue University, West Lafayette, IN 47907, USA; lalanin@purdue.edu; 2West Lafayette Junior/Senior High School, West Lafayette, IN 47906, USA; xuzihan2022@gmail.com

**Keywords:** gender difference, psychosocial distress, social support, financial well-being, informal caregivers, older adults

## Abstract

**Background**: Financial stress can lead to emotional and psychosocial distress among informal caregivers of older adults and can have a profound impact on their overall well-being. While social support may buffer financial stress, the role of gender in moderating these relationships is less understood. This study examined whether gender moderates the associations between psychosocial distress, social support, and financial well-being among informal caregivers of older adults. **Methods**: A cross-sectional survey was conducted between December 2023 and March 2024 among 589 informal caregivers of older adults residing in twelve rural states in the North Central United States. After applying listwise deletion for missing data, 533 caregivers with complete responses were retained for the final analysis. Financial well-being was measured using the Consumer Financial Protection Bureau (CFPB) Financial Well-Being Scale, psychosocial distress with a five-item distress scale, and social support with the OSSS-3. Gender, sociodemographic characteristics, and caregiving status were also collected. Independent t-tests, ANOVA, Pearson correlations, and multivariable linear regressions with interaction terms were used to assess the relationships among the study variables. **Results**: Male caregivers reported significantly higher financial well-being than females (52.66 vs. 50.12, *p* = 0.036). Stronger social support was associated with greater financial well-being (mean difference = 11.80, *p* < 0.001). Psychosocial distress was negatively correlated with financial well-being (r = –0.49, *p* < 0.001). Regression analyses revealed significant gender moderation: distress reduced financial well-being more sharply for males, while strong social support benefited females more substantially than males. Older age, higher income, and better self-rated health were also positively associated with financial well-being. **Conclusions**: Findings highlight gender-specific patterns in how psychosocial distress and social support influence financial well-being among informal caregivers. Future programs should consider gender-specific needs while addressing psychosocial and financial well-being of informal caregivers of older adults.

## 1. Introduction

Informal caregivers of older adults face substantial financial challenges that extend beyond the clinical demands of care. Financial burden remains one of the most pervasive stressors affecting caregivers in both home and institutional settings [1,2]. This burden encompasses direct and indirect caregiving costs, including medical expenses, time devoted to care, and expanded roles and responsibilities [3,4]. In addition to healthcare-related costs, caregivers often incur housing, transportation, and other ancillary expenses, which together can exceed $7000 annually [5]. On average, family caregivers spend approximately 26% of their income on caregiving-related expenses, and nearly half rely on personal funds to meet household needs associated with caregiving [6,7,8]. Among informal caregivers who provide unpaid care, financial strain is especially pronounced, with more than half reporting dissatisfaction with household income and nearly one-third struggling to meet basic needs [9].

Beyond its economic implications, financial strain frequently undermines caregivers’ overall economic stability. Many caregivers report reducing savings, delaying bill payments, accumulating debt, and limiting spending on essential needs as they attempt to balance caregiving responsibilities with household obligations [3,4,10]. These challenges are often compounded by limited financial literacy and insufficient guidance from service providers, which can restrict caregivers’ awareness of and access to available financial support [3,7,11]. Consequently, financial hardship among caregivers is closely linked to employment disruptions, reduced work hours, career sacrifices, and strain on family roles, factors that collectively heighten economic vulnerability over time [11,12].

These financial pressures are further intensified for caregivers residing in rural areas. Structural barriers such as limited availability of healthcare facilities, long travel distances for medical appointments, higher transportation costs, and fewer local support services can substantially increase both the financial and emotional demands of caregiving [13,14]. In addition, rural caregivers often face reduced economic opportunities, limited access to employer-sponsored benefits, and fewer respite care options, all of which compound caregiving-related financial strain [15]. As a result, rural caregivers represent a particularly vulnerable population whose experiences may differ in important ways from those of their urban counterparts.

Financial strain among caregivers has well-documented consequences for psychological and physical well-being. Higher levels of financial difficulty are consistently associated with increased psychosocial distress, poorer self-rated health, and declines in overall quality of life [9,11,16,17]. These associations are further shaped by demographic factors. Older caregivers (aged 65 years and above) often report higher financial well-being than younger caregivers, potentially reflecting accumulated assets, retirement income, and fewer competing financial demands. In contrast, younger caregivers, individuals with lower socioeconomic status, and male caregivers frequently experience greater financial strain [18,19,20]. Young adult caregivers, in particular, report heightened emotional distress and financial insecurity compared to non-caregiving peers [16]. At the same time, a substantial proportion of caregivers are older adults managing their own health challenges while providing care to others, further complicating their financial and emotional circumstances [12].

Gender differences provide an important lens through which to understand these patterns. Women constitute the majority of informal caregivers and typically devote more hours to caregiving than men, increasing their exposure to financial strain and caregiving-related stress [12,21]. Gender-role theory suggests that men and women internalize distinct societal expectations that influence their responses to stress and caregiving demands [22]. Women are often socialized to be nurturing and emotionally expressive, whereas men are encouraged to be self-reliant and less inclined to seek help. Socio-behavioral models further indicate that gendered norms shape coping strategies, perceptions of vulnerability, and help-seeking behaviors [23,24]. As a result, women tend to draw more heavily on social and emotional support, while men may be less likely to disclose distress or seek financial assistance potentially exacerbating stress when financial pressures intensify [25,26].

Despite extensive documentation of financial strain, psychosocial distress, and social support among informal caregivers, important gaps remain in the literature. Existing studies have often examined these factors independently, with limited attention to how they interact or whether their associations differ by gender [27]. Moreover, rural caregivers remain underrepresented in national caregiving research, despite facing distinct structural barriers and resource constraints that may shape financial well-being in unique ways. Addressing these gaps is essential for developing interventions that are both contextually appropriate and responsive to caregivers lived experiences.

Accordingly, this study examines the associations among psychosocial distress, social support, and financial well-being among rural informal caregivers of older adults, with a specific focus on the moderating role of gender. By identifying gender-specific patterns within these relationships, this research aims to inform the development of gender-responsive interventions, targeted financial literacy initiatives, and policies designed to support informal caregivers in rural communities.

### Purpose

This study aims to investigate whether gender moderates the relationships between (1) psychosocial distress and financial well-being, and (2) social support and financial well-being.

## 2. Methods

### 2.1. Study Design

A cross-sectional survey design was employed to examine the financial well-being of rural informal caregivers in the north-central regions of the United States. Data were collected using a structured questionnaire from validated and standardized instruments, including the CFPB Financial Well-Being Scale, the OSSS-3 Social Support Scale, and a psychosocial distress index derived from commonly used stress, anxiety, depression, and loneliness measures using standardized tools hosted on Qualtrics software. Qualtrics software was selected to develop and administer the survey online among the participants. The platform ensured compliance with Purdue University’s IRB and data-protection requirements, including secure storage, encrypted transmission, and anonymous participation features essential for collecting sensitive information related to financial well-being, psychosocial distress, and caregiving experiences.

### 2.2. Setting and Sample

A cross-sectional survey was designed using a convenience sampling approach to collect data across twelve states in the North Central rural regions of US, including Illinois, Indiana, Iowa, Kansas, Michigan, Minnesota, Missouri, Nebraska, North Dakota, Ohio, South Dakota, and Wisconsin. The study was conducted between December 2023 and March 2024. A cross-sectional convenience sampling approach was used, as participation remained voluntary and relied on the’ willingness and ability to respond to a web-based survey. Therefore, while these strategies enhanced geographic diversity and improved access to hard-to-reach rural caregivers, the sample does not constitute a statistically representative sample of all rural caregivers.

### 2.3. Participant Eligibility

Eligible participants were 18 years or older and currently providing unpaid or informal care to an older adult aged 60 or above with a chronic illness or disability residing at home, in hospice, or within a residential care facility.

### 2.4. Recruitment Procedures

Participants were recruited using a multimethod approach designed to reach caregivers across rural communities. Recruitment was conducted in partnership with rural hospitals, and clinics, which assisted in distributing study information to potential participants. Flyers were posted in public locations such as libraries, churches, community centers, and rural health facilities. In addition, members of the research team engaged in in-person outreach at rural community fairs and local events to increase visibility and reach caregivers who might not be connected to formal service networks.

To further diversify the sample, a professional survey panel company was contracted to disseminate the study invitation to eligible caregivers residing in rural regions. Recruitment materials were also shared through the social media pages and newsletters of caregiving and aging-related organizations. The survey required approximately 15–20 min to complete.

### 2.5. Study Measures

#### 2.5.1. Dependent Variable

**Financial Well-Being**. Perceived financial well- being was assessed using five-item scale Consumer Financial Protection Bureau’s [28]. This validated scale captures individuals’ perceptions of their financial security and control (Appendix A). Participants responded to five items: (1) Because of my money situation, I feel like I will never have the things I want in life; (2) I am just getting by financially; (3) I am concerned that the money I have or will save won’t last; (4) I have money left over at the end of the month; and (5) My finances control my life. Items were rated on a 5-point Likert scale, with negatively worded items (Items 1–3, and 5) reverse-coded. Total raw scores were then converted into a standardized financial well-being score ranging from 19 to 90 (Appendix A), based on respondents’ age group (18–61 years or 62+ years) and the method of survey administration (self-administered or administered by someone else). Higher scores indicate greater financial well-being. The Cronbach’s alpha was 0.805.

#### 2.5.2. Independent Variables

**Social Support**. Perceived social support was measured using the Oslo-3 Social Support Scale (OSSS-3; Appendix A), a brief, validated three-item instrument designed to assess the level of social support [29]. Total scores range from 3 to 14, with higher scores indicating stronger perceived social support. Participants were categorized into three groups based on their OSSS-3 scores: 3–8 (poor social support), 9–11 (moderate social support), and 12–14 (strong social support). The OSSS-3 has been widely used in research and has demonstrated acceptable reliability and construct validity [29]. Cronbach’s alpha was 0.488 in the current study. Given that the OSSS-3 contains only three items, this level of internal consistency is not unexpected. Although the Cronbach’s alpha for the OSSS-3 in this study was relatively low (α = 0.488), this does not indicate that the instrument lacks validity or reliability. The OSSS-3 is a standardized, widely used, and psychometrically supported tool designed intentionally as a three-item scale, and short scales commonly produce lower alpha coefficients due to limited item redundancy. Prior research, including the original validation by Kocalevent et al., has demonstrated that the OSSS-3 possesses good construct validity, strong predictive utility, and acceptable reliability for population-level research, even when internal consistency values appear modest. Additionally, all instruments used in this study including the CFPB Financial Well-Being Scale and the psychosocial distress index are standardized measures with established validity in diverse populations. The use of these validated tools, combined with rigorous data-collection procedures through the Qualtrics platform, supports the overall reliability and methodological soundness of the study.

**Psychosocial Distress**. Participants’ psychosocial distress was assessed using five self-reported items: feeling stressed, anxious, depressed, lonely, and having no time for oneself (Appendix A). Each item was measured on a scale from 0 to 10, with 0 indicating the lowest level of distress and 10 indicating the highest level of distress. A total distress score was calculated by summing the responses across all five items, with higher scores indicating greater psychosocial distress. Cronbach’s alpha was 0.847 in the current study.

**Gender.** Participants were categorized into two gender groups: male and female. The dataset did not include individuals identifying as gender-fluid or non-binary.

#### 2.5.3. Covariates

In addition to the variables described above, the study also collected demographic and background information, including caregivers’ age (18–64 years or 65+ years), race and ethnicity (non-Hispanic White or non-White), education level (high school or less, some college, or college degree or higher), marital status (married/partnered, never married, or divorced/widowed/separated), and annual household income (less than $20,000; $20,000–$49,999; $50,000–$74,999; or ≥ $75,000). Participants were also asked whether they provided care for older adults (yes or no) and to rate their overall health (poor, fair, good, or very good).

### 2.6. Statistical Analysis

Descriptive statistics were used to summarize sample characteristics. Means and standard deviations were reported for continuous variables, while frequencies and percentages were used for categorical variables. An independent samples t-test was conducted to examine differences in financial well-being by gender, and a one-way analysis of variance (ANOVA) was used to assess differences in financial well-being across levels of social support. Pearson correlation coefficients were calculated to evaluate the association between psychosocial distress and financial well-being. To explore whether the associations between psychosocial distress and financial well-being, as well as between social support and financial well-being, differed by gender, a linear regression model was conducted including two interaction terms: psychosocial distress × gender and social support × gender. All regression models controlled for age, race/ethnicity, education level, marital status, annual household income, caregiving status for older adults, and self-rated health. Multicollinearity was not a concern in the linear regression analysis, as the variance inflation factor (VIF) for all variables was less than 3. All statistical analyses were conducted using Stata version 19.0 (StataCorp LLC, College Station, TX, USA), with a *p*-value of less than 0.05 considered statistically significant.

### 2.7. Ethical Considerations

Ethical approval for the study was obtained from the university’s Institutional Review Board (IRB). Informed consent was embedded within the online Qualtrics survey, which participants completed prior to beginning the questionnaire. To ensure confidentiality and anonymity, all surveys were coded numerically without personal identifiers. Data were securely stored on the Qualtrics platform and accessed only by the research team, following institutional ethical guidelines for data protection.

## 3. Results

A total of (n= 589) participants completed the surveys. Data showed 56 participants (9.51%) had missing data. Listwise deletion was employed to handle the missing data. Data from n = 533 participants (90.49%) with complete responses were included in the final analysis.

Table 1 presents the sample characteristics of n = 533 caregivers. The majority (70.73%) were aged 18–64 years, while 29.27% were older adult caregivers (65 years and older). Approximately 45.97% of the sample were female caregivers, and the majority (88.56%) identified as non-Hispanic White. Educational attainment was relatively evenly distributed: 34.15% had a high school diploma or less, 34.15% had some college education, and 31.71% held a college degree or higher. More than half of the caregivers (56.29%) were married or partnered; 23.26% had never married, and 20.45% were divorced, widowed, or separated. In terms of annual household income, 17.82% reported earning less than $20,000, 34.52% earned between $20,000 and $49,999, 21.58% between $50,000 and $74,999, and 26.08% reported incomes of $75,000 or more. About 36.77% of caregivers provided care for older adults. Regarding self-rated health, 5.82% rated their health as poor, 24.95% as fair, 53.85% as good, and 15.38% as very good. Nearly half of caregivers (48.22%) reported low levels of perceived social support, 43.71% reported moderate support, and only 8.07% reported strong support. The average psychosocial distress score was 17.90 (range: 0–50), and the mean financial well-being score was 51.49 (range: 19–90).

As shown in Table 2, male caregivers reported significantly higher financial well-being than female caregivers, with a mean difference of 2.54 points (males: 52.66; females: 50.12; 95% CI: 0.16–4.92; *p* = 0.036). ANOVA results also indicated significant differences in financial well-being across levels of social support (*p* < 0.001; Table 2). Post hoc analysis revealed that caregivers who reported strong and moderate social support had significantly higher financial well-being scores, 11.80 points (95% CI: 6.61–16.98; *p* < 0.001) and 7.04 points (95% CI: 4.19–9.89; *p* < 0.001), respectively, compared to those with poor social support. Although caregivers who reported strong social support had 4.76 points higher financial well-being than those with moderate support, this difference was not statistically significant (*p* = 0.099). As expected, there was a moderate negative correlation between psychosocial distress and financial well-being (r = –0.491, *p* < 0.001), indicating that higher levels of distress were associated with lower financial well-being.

As shown in Table 3 and Figure 1, psychosocial distress was negatively associated with financial well-being, even after adjusting for demographic and other potential confounding variables. Moreover, the interaction term between psychosocial distress and gender was statistically significant (β = 0.26, *p* = 0.002; Table 3) indicating that gender moderates the relationship between psychosocial distress and financial well-being. Specifically, as psychosocial distress increased, financial well-being declined more sharply among males than females (Figure 1). At lower levels of psychosocial distress (scores below 30), males reported higher financial well-being compared to females; however, at higher levels of distress (scores above 30), this pattern reversed, with males reporting lower financial well-being than females (Figure 1).

For the relationship between social support and financial well-being, we found strong interaction between strong social support and gender (β = 10.64, *p* = 0.006; Table 3), indicating that gender moderates this association. Among caregivers reporting poor or moderate social support, females reported lower financial well-being than males (Figure 1). However, when females perceived strong social support, their financial well-being increased substantially, surpassing that of males with similarly strong social support, although the gender difference at this level was not statistically significant (Figure 1).

Among the covariates, caregivers aged 65 years and older reported significantly better financial well-being compared to those aged 18–64 years (β = 7.78, *p* < 0.001; Table 3). Higher annual household income was also associated with greater financial well-being: caregivers with $50,000–$74,999 had significantly higher scores than those earning less than $20,000 (β = 3.56, *p* = 0.035), as did those with incomes of $75,000 or more (β = 5.33, *p* = 0.002; Table 3). In addition, caregivers who rated their health as very good reported better financial well-being compared to those with poor or fair self-rated health (β = 3.26, *p* = 0.048; Table 3). Caregivers who reported good health also had higher financial well-being scores than those with poor or fair health, although this difference was not statistically significant (β = 0.82, *p* = 0.476; Table 3). Race/ethnicity, education level, marital status, and caregiving for an older adult were not significantly associated with financial well-being (Table 3).

## 4. Discussion

This study examined the associations among psychosocial distress, social support, and financial well-being, and the moderating role of gender in these relationships among a multi-state sample of rural informal caregivers. Consistent with prior research, psychosocial distress was moderately and negatively associated with financial well-being, indicating that caregivers reporting higher distress also reported lower financial well-being. Because of the cross-sectional nature of the data, these findings should be interpreted as associational rather than causal, and no conclusions can be drawn about the directionality of these relationships.

Although the present study did not directly assess employment disruptions, caregiving intensity, or hours of care factors commonly linked to caregiver distress existing literature suggests that these mechanisms may co-occur with both psychological strain and financial challenges [30,31,32,33]. In rural contexts, caregivers often face additional structural constraints, including long travel distances for care, limited broadband access, and fewer locally available services [9,11,13,14,15]. These contextual factors may be associated with heightened distress and financial strain, though they were not directly measured in this study. Conversely, rural caregivers frequently rely on informal support from family members, neighbors, churches, and community organizations, which may partially compensate for limited formal infrastructure [33,34]. Our findings therefore highlight the potential importance of informal support systems, while acknowledging that their role cannot be interpreted as protective in a causal sense.

A key contribution of this study is the identification of gender as a moderator, rather than a mediator, of the relationship between psychosocial distress and financial well-being. As distress increased, male caregivers exhibited a steeper decline in financial well-being compared to female caregivers. Prior studies suggest that men may be less likely to engage in help-seeking behaviors or utilize diverse coping strategies, whereas women more frequently draw on social networks such as family and friends [24,25,26,32,33,35,36]. While these explanatory mechanisms were not directly measured, the observed interaction indicates that the association between distress and financial well-being differs by gender, warranting further investigation in longitudinal or mixed-methods studies. These findings suggest that gender-responsive interventions aimed at strengthening coping resources may be particularly relevant for male caregivers, though causal inferences cannot be made.

Gender moderated the association between social support and financial well-being. Women reported lower financial well-being when support was poor or moderate, whereas this difference narrowed and appeared to reverse when support was strong. However, this interaction did not reach statistical significance, and the small size of the strong-support subgroup necessitates cautious interpretation. Rather than indicating a definitive gender effect, this pattern may reflect caregiving norms in rural communities, where women often play central roles in organizing informal care networks and accessing community resources [37]. These findings point to the potential relevance of gender-sensitive and rural-specific approaches to strengthening social support among caregivers experiencing financial strain.

Across the full sample, social support was positively associated with financial well-being, with caregivers reporting moderate or strong support also reporting higher financial well-being. This finding aligns with prior research indicating that social networks are associated with improved emotional, social, and financial outcomes [30,31,38]. However, social support was measured using a brief three-item scale, which may not fully capture the multidimensional nature of caregivers’ support systems. As such, the magnitude of these associations should be interpreted cautiously. Future research would benefit from using more comprehensive measures of social support to improve reliability and analytic precision.

Finally, age and self-rated health were associated with financial well-being. Older caregivers reported higher financial well-being than younger caregivers, possibly reflecting accumulated assets, savings, or reduced competing financial demands. Caregivers reporting very good health also reported higher financial well-being than those in poorer health, underscoring the close interrelationship between health and financial status. However, given the cross-sectional design, it is not possible to determine whether better health contributes to financial well-being, whether financial security supports better health, or whether both are influenced by unmeasured factors [34].

### 4.1. Strengths and Limitations

#### 4.1.1. Strengths

Our study is one of the few studies to examine gender patterns and their associations among psychosocial distress, social support, financial well-being, and gender particularly among rural informal caregivers of older adults, a hard-to-reach population. The data gathered across twelve rural states in US adds to the geographic diversity and meaningful variations in the findings. Use of validated scales. The study contributes new knowledge by testing gender moderation effects, offering insights that extend beyond traditional main-effect models in rural caregiving research.

#### 4.1.2. Limitations

The particular study design prevents us from drawing any causal inferences among the study variables. Longitudinal studies are therefore recommended to establish any temporal relationships and potential bidirectional effects. The reliance on self-reported data may have introduced recall and social desirability biases, particularly in sensitive areas such as financial well-being and emotional distress. The sample was drawn primarily from rural caregivers and may not be generalizable to other populations. Web-based surveys facilitate participation among geographically dispersed rural caregivers; constraints such as self-selection and internet access remain limiting full representation. The measurement of social support contained only three items and may not have fully captured the nuances of caregivers’ support networks. The survey did not assess whether caregivers were responsible for more than one older adult, that might be a limitation in the findings. Some participants may have been providing care to multiple individuals; an arrangement associated with disproportionately higher financial strain. Future studies should incorporate measures of caregiving load, including the number of care recipients, to more accurately evaluate its impact on caregiver financial and emotional outcomes.

## 5. Conclusions

Rural caregiving brings several financial burdens and influences informal caregivers’ health and well-being of both genders in different ways. Tailored interventions are needed to address the emotional and financial well-being needs of male and female informal caregivers of older adults in rural and underserved areas.

## 6. Practical Implications for Research, Practice, Policy, and Health Education

There are several implications. Future longitudinal and mixed-methods studies are needed to better understand the causal pathways linking psychosocial distress, social support, and financial well-being across diverse geographic and ethnic populations. Future qualitative inquiry into caregivers lived experiences could illuminate underlying mechanisms and barriers affecting financial health and resilience among caregivers in rural areas. From a practice and policy standpoint, this study underscores an urgent need for health systems, community organizations, and policymakers to develop programs that could address the specific needs of financial health and planning across both males and females. Providers should initiate early conversations about financial planning, insurance literacy, and social support to support informal caregivers’ well-being. Health education programs should provide information about financial resources and programs.

## Figures and Tables

**Figure 1 healthcare-14-00017-f001:**
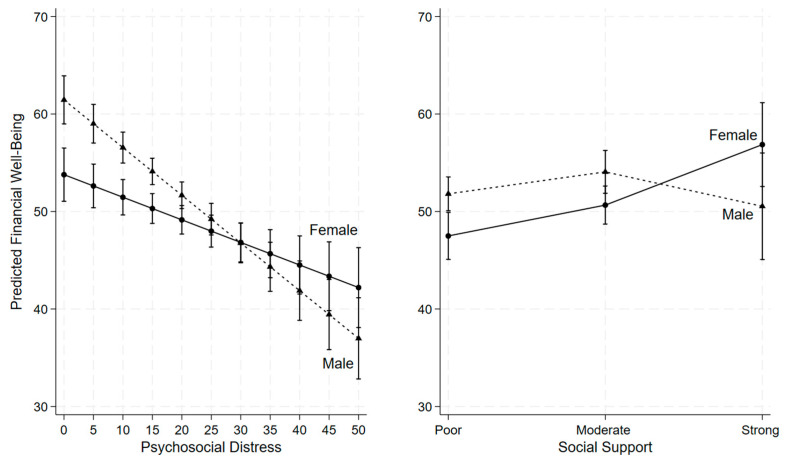
Gender differences in the associations of financial well-being with psychosocial distress and social support (n = 533).

**Table 1 healthcare-14-00017-t001:** Sample characteristics (n = 533).

Variables	N (%) or Mean ± SD
Age group	
18–64 years	377 (70.73%)
65+ years	156 (29.27%)
Gender	
Male	288 (54.03%)
Female	245 (45.97%)
Race and ethnicity	
Non-Hispanic White	472 (88.56%)
Non-White	61 (11.44%)
Education level	
≤High school	182 (34.15%)
Some college	182 (34.15%)
College degree or higher	169 (31.71%)
Marital status	
Married/Partnered	300 (56.29%)
Never married	124 (23.26%)
Divorced/Widowed/Separated	109 (20.45%)
Annual household income	
Less than $20,000	95 (17.82%)
$20,000 to $49,999	184 (34.52%)
$50,000 to $74,999	115 (21.58%)
≥$75,000	139 (26.08%)
Caregiving status for older adults	
Yes	196 (36.77%)
No	337 (63.23%)
Self-rated health	
Poor	31 (5.82%)
Fair	133 (24.95%)
Good	287 (53.85%)
Very Good	82 (15.38%)
Social support	
Poor support	257 (48.22%)
Moderate support	233 (43.71%)
Strong support	43 (8.07%)
Psychosocial distress	17.90 ± 12.16
Financial well-being	51.49 ± 13.98

Note: SD: standard deviation.

**Table 2 healthcare-14-00017-t002:** Differences in financial well-being by gender and levels of social support (n = 533).

	N	Mean ± Standard Deviation	*p* Value
Gender			
Male	288	52.66 ± 13.83	**0.036**
Female	245	50.12 ± 14.06	
Social support			
Poor support	257	47.46 ± 13.47	**<0.001**
Moderate support	233	54.50 ± 12.91	
Strong support	43	59.26 ± 15.44	

**Table 3 healthcare-14-00017-t003:** Results from the linear regression model examining the main and interaction effects of psychosocial distress, social support, and gender on financial well-being (n = 533).

Outcome: Financial Well-Being	B	*p* Value	95% CI	
			Lower	Upper
Gender (ref. Male)				
Female	−8.93 *	<0.001	−13.54	−4.31
Psychosocial distress	−0.49 *	<0.001	−0.61	−0.37
Psychosocial distress × Gender	0.26 *	0.002	0.09	0.42
Social support (ref. Poor support)				
Moderate support	2.25	0.120	−0.59	5.10
Strong support	−1.28	0.665	−7.05	4.50
Social support × Gender (ref. Poor support, Male)				
Moderate support, Female	0.91	0.670	−3.27	5.09
Strong support, Female	10.64 *	0.006	3.05	18.23
Age group (ref. 18–64 years)				
65+ years	7.78 *	<0.001	5.45	10.12
Race and ethnicity (ref. Non-White)				
Non-Hispanic White	−2.62	0.094	−5.70	0.45
Education level (ref. ≤ High school)				
Some college	−0.51	0.665	−2.85	1.82
College degree or higher	1.94	0.134	−0.60	4.48
Annual household income (ref. < $20,000)				
$20,000 to $49,999	1.68	0.255	−1.21	4.56
$50,000 to $74,999	3.56 *	0.035	0.26	6.86
≥ $75,000	5.33 *	0.002	1.89	8.78
Marital status (ref. Married/Partnered)				
Never married	−1.95	0.135	−4.51	0.61
Divorced/Widowed/Separated	−2.41	0.074	−5.06	0.23
Caregiving status for older adults (ref. No)				
Yes	1.21	0.243	−0.82	3.24
Self-rated health (ref. Poor/Fair)				
Good	0.82	0.476	−1.43	3.06
Very good	3.26 *	0.048	0.03	6.50

Notes: * *p* < 0.005--CI: confidence interval. Psychosocial distress × Gender represents the interaction term between psychosocial distress and gender. Social support × Gender represents the interaction term between levels of social support and gender.

## Data Availability

The original contributions presented in this study are included in the article/Appendix A. Further inquiries can be directed at the corresponding author.

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
