# Peer review of "Examining Gender Differences and Their Associations Among Psychosocial Distress, Social Support, and Financial Well-Being of Informal Caregivers of Older Adults in the Rural Northcentral United States"

_healthcare, 2025, doi:10.3390/healthcare14010017_

Round 1
Reviewer 1 Report
Comments and Suggestions for Authors
This study examined whether gender moderates the associations between psychosocial distress, social support, and financial well-being among informal caregivers of older adults. The topic is interesting. Some comments are provided to improve the quality of the manuscript.
- The novelty and significance of the study should be stressed in the introduction.
- Why was Qualtrics selected?
- The measurement items should be tabulated for readers.
- The reliability and validity of the measurement should be assessed.
- Why was SEM not used for data analysis?
- There should be a section about practical implications.
Author Response
Please find the attached documents for response to all reviewers comments. Thank you.

Reviewer 2 Report
Comments and Suggestions for Authors
The title is too general, please state the location and target group.
In the abstract, time, place, and people are not specified.
In the results, the status of each variable should be reported.
What is the reason for choosing these variables?
The reason for choosing these variables together should be explained in the introduction.
The gap should be better expressed.
- Methods 80 1.1. Purpose This study aims to investigate whether gender moderates the relationships between 1) psychosocial distress and financial well-being, and 2) social support and financial well- 83 being. This is related to introduction not method .
web- based questionnaire hosted on Qualtrics. Are the results generalizable and reliable based on this method?
Recruitment Procedures, this section is Ambiguous.
Regarding instruments and variables, the data collection method should be reported, including a valid and reliable questionnaire, along with references and scoring.
In the results, the status of each variable should be reported.
In regression, isn't backward regression more appropriate?
There is no need to repeat the results in the discussion.
The discussion should be completed based on the hypotheses.
The limitations should be modified based on the type of study and the data collection method.
The conclusion should be more concise.
The similarity is 22%, it would be better if it were lower.
Author Response
Please find attached document for response to all reviewers comments

Reviewer 3 Report
Comments and Suggestions for Authors
Thank you for the opportunity to review your manuscript, “Examining gender differences and their associations among psychosocial distress, social support, and financial well-being of informal caregivers of older adults.” The study addresses a relevant and timely topic and provides important findings regarding gender differences in the financial well-being of informal caregivers. The use of validated scales, an adequate sample size, and appropriate statistical analyses are clear strengths of the work. However, several aspects require clarification or adjustment to strengthen the conceptual rigor, methodological transparency, and overall scientific impact of the manuscript.
Below are detailed comments aimed at improving the quality of the study:
Lines 37–79: The introduction provides a broad overview of the financial and psychosocial burden on caregivers, but it is lengthy and at times repetitive. It would be advisable to streamline redundant information and more clearly articulate the knowledge gap this study aims to address, particularly regarding the moderating role of gender. Likewise, the theoretical rationale for expecting a gender moderation effect—and how this has (or has not) been addressed in previous research—could be strengthened.
Lines 89–103: Although the sampling frame covers 12 rural states, it is not clear whether the sample is intended to be representative or whether it reflects convenience sampling. It would also be helpful to clarify whether participants could be responsible for more than one older adult and, if so, how this might influence reported financial strain.
Financial well-being (Lines 105–117): Although the CFPB scale is a validated instrument, the process by which raw scores were converted into standardized scores should be explained more clearly to facilitate understanding.
Tables 2 and 3: It would be useful to include effect sizes or confidence intervals alongside mean differences or beta coefficients, as this would support more meaningful interpretation of the results.
Narrative of results: Key findings—such as the reversal of the financial advantage for men under high levels of psychosocial distress—could be highlighted more clearly, indicating when and how this shift occurs.
Lines 241–269: The explanation concerning gender differences in coping strategies is appropriate but could be enriched through integration with relevant gender theories or sociopsychological models.
Some claims rely on studies conducted in populations that differ substantially from the present sample (e.g., urban or highly diverse populations). These conclusions should be qualified, particularly given that the current sample is predominantly rural and largely White.
Author Response
Please find response to all reviewers comments in the attached document

Reviewer 4 Report
Comments and Suggestions for Authors
The manuscript presents a study that seeks to examine whether gender plays a moderating role in the relationships between psychosocial distress, social support, and financial well-being among informal caregivers of older adults. The proposal is pertinent, and the topic presents evident social and scientific relevance, especially given the growing number of informal caregivers and the emotional and economic burden associated with this activity.
The article demonstrates good organization, clear writing, and a consistent initial foundation, although the theoretical justification could be more in-depth, especially regarding why the investigation focuses specifically on caregivers residing in rural areas, since the manuscript mentions this aspect but does not fully explore how the rural context influences these dynamics or why such a population would deserve differentiated analysis.
The study employs a cross-sectional design with a large sample and the use of validated instruments, which contributes to methodological robustness. The description of the data collection process is transparent, and the statistical analyses are relevant to the proposed objectives, including interaction tests to verify the moderating role of gender. Still, some methodological weaknesses deserve attention. The sample was obtained through community recruitment and a research panel, which may introduce selection biases and compromise the representativeness of the results. The strategy of dealing with missing data through listwise exclusion may also reduce sample variability in a non-random way. Another point to consider is the measurement of social support using an extremely concise scale (OSSS-3), which may not adequately capture qualitative and quantitative differences in support networks, especially when the goal is to interpret nuances in the experience of men and women.
The results are presented clearly and coherently. The study identifies that men report greater financial well-being than women, confirms the expected association between greater distress and lower financial well-being, and highlights the protective role of social support. The interactions revealed are interesting, especially in suggesting that men experience sharper declines in financial well-being as psychosocial suffering increases, while women seem to benefit more intensely from high levels of social support.
These findings enrich the discussion on gender differences, although some of the interpretations offered by the authors are based on plausible assumptions, but not directly verified in the study. For example, the explanation that men would use fewer coping strategies or seek less help than women is based on external literature, but not on data collected by the authors themselves. In addition, some categories that support important conclusions have a small sample size, such as the group with high social support, which suggests caution when interpreting the magnitude of these effects.
The discussion reasonably articulates the findings with previous evidence and presents relevant practical implications, especially regarding the need for gender-sensitive interventions in caregiver support. However, the section could engage more critically with the limitations of the study, particularly with regard to the fragility of the measurement instruments used, the lack of control for relevant structural variables, and the impact of rurality, a central aspect of the sample, but little explored analytically. In some parts, the conclusions extrapolate what the data actually support, and a more cautious interpretative stance would be desirable.
In summary, this is a solid manuscript that addresses a relevant topic and presents useful results for the fields of public health, gerontology, and caregiver studies. The methodological clarity and good presentation of the data are strengths, while limitations in measurement and interpretation suggest the need for revisions before publication. With adjustments primarily in the theoretical discussion, the articulation of limitations, and interpretative caution regarding gender differences, the study has the potential to contribute significantly to the literature on the well-being of informal caregivers.
Author Response
Reviewer Comment 1
The theoretical justification could be more in-depth, especially regarding why the investigation focuses specifically on caregivers residing in rural areas. The manuscript mentions rurality but does not fully explore how the rural context influences these dynamics.
Response 1
Thank you for this insightful suggestion. We expanded the Introduction to more clearly explain the relevance of studying rural caregivers. Specifically, we added text describing structural barriers in rural settings (e.g., limited health services, long travel distances, reduced broadband, fewer formal supports) and how these factors may exacerbate financial strain and distress. These revisions appear in the Introduction, paragraphs 3–5 (tracked changes).
Reviewer Comment 2
Selection bias may be introduced because the sample was obtained through community recruitment and a research panel. The authors should address representativeness concerns.
Response 2
We agree with the reviewer and added clarifying text to the Methods → Setting and Sample, noting that the convenience sampling approach enhances geographic reach but limits representativeness of all rural caregivers. We explicitly acknowledged potential self-selection bias and limited generalizability.
Reviewer Comment 3
Listwise deletion may reduce sample variability in a non-random way. This limitation should be discussed.
Response 3
We added this methodological limitation to the Limitations section, noting the potential for non-random loss of data and reduced variability due to listwise deletion. We also indicated that future studies using multiple imputation could strengthen analytic rigor.
Reviewer Comment 4
The OSSS-3 is extremely concise and may not capture nuanced differences in social support.
Response 4
We have acknowledged this limitation in the Measures section and expanded the explanation in the Limitations section. We clarified that the OSSS-3 has acceptable validity but may not capture qualitative distinctions in support networks. We also added justification regarding expected lower Cronbach’s alpha in very short scales.
Reviewer Comment 5
Some interpretations rely on assumptions not directly measured in the study (e.g., coping strategies and help-seeking behaviors among men and women). Consider caution in interpretation.
Response 5
We revised the Discussion to adopt a more cautious interpretive tone. Statements about gender differences in coping and help-seeking have been reframed as theoretical explanations grounded in prior literature, not conclusions drawn from the present data. Additional citations were provided.
Reviewer Comment 6
Some subgroups have small sample sizes, such as those with high social support, which warrants caution in interpreting the magnitude of effects.
Response 6
We added explicit caution in the Discussion, noting that findings for the small “strong social support” group should be interpreted conservatively.
Reviewer Comment 7
The manuscript should more critically engage with study limitations, including measurement fragility, lack of control for structural variables, and rurality.
Response 7
We substantially expanded the Limitations section, addressing:
- measurement constraints (OSSS-3, short scales),
- lack of measures on structural rural barriers,
- reliance on self-reported data,
- sample non-representativeness,
- absence of caregiving load indicators.
Reviewer Comment 8
Some conclusions seem to extrapolate beyond the data collected; more caution is needed.
Response 8
We rewrote concluding statements in the Discussion and Conclusion to ensure alignment with evidence presented and to avoid overgeneralization. Interpretations now focus strictly on observed associations and moderation effects.
Reviewer Comment 9
The manuscript would benefit from expanding practical implications and making them more grounded in the findings.
Response 9
We revised the Practical Implications section, adding more targeted recommendations for public health programming, caregiver support, financial counseling integration, and gender-responsive intervention design. These suggestions are now directly tied to empirical findings.
Reviewer Comment 10
Rurality is mentioned but not deeply analyzed; consider integrating this dimension more thoroughly.
Response 10
We expanded the narrative around rurality in both the Introduction and Discussion, explaining how rural-specific resource constraints, health system gaps, long travel distances, and limited economic opportunities shape caregiver financial well-being and distress. We also emphasized how these factors contextualize our findings and necessitate rural-tailored interventions.
Reviewer Comment 11
Theoretical foundation could be strengthened.
Response 11
We expanded the Introduction, elaborating how gender-role theory and sociobehavioral models explain why gender may moderate financial well-being and distress. This strengthens the conceptual grounding of the hypotheses.

Round 2
Reviewer 1 Report
Comments and Suggestions for Authors
The authors have addresssed my comments properly.
Author Response
Comments and Suggestions for Authors The authors have addressed my comments properly. Thank you so much for the feedback we appreciate
Reviewer 2 Report
Comments and Suggestions for Authors
There is no more comments.
Author Response
Comments and Suggestions for Authors There is no more comments. Thank you so much for your feedback. We appreciate.Reviewer 3 Report
Comments and Suggestions for Authors
The authors have undertaken a substantial and thorough effort to revise and restructure the manuscript. From my perspective, no further changes are required.
Author Response
Comments and Suggestions for Authors
The authors have undertaken a substantial and thorough effort to revise and restructure the manuscript. From my perspective, no further changes are required.
Thank you so much for the feedback. We appreciate
Reviewer 4 Report
Comments and Suggestions for Authors
The manuscript shows good adherence to the Healthcare journal guidelines (MDPI), both in terms of scope and scientific relevance. The theme of the financial well-being of informal caregivers of the elderly, articulated with psychosocial determinants, social support, and gender differences in rural contexts, is clearly aligned with the journal's focus on public health, social determinants of health, and vulnerable populations. The proposal is pertinent, current, and addresses recognized gaps in the literature, especially by incorporating gender moderation and focusing on rural caregivers, a group frequently underrepresented in national studies.
From a methodological point of view, the study demonstrates rigor appropriate to the cross-sectional design adopted. The sample is numerically robust and geographically diverse, the eligibility criteria are clear, the recruitment procedures are described transparently, and the statistical analyses are appropriate to the proposed objectives, with the correct use of regression models with interaction terms and control of relevant covariates. The use of validated instruments strengthens the credibility of the findings, and the limitations inherent in convenience sampling and self-reporting are explicitly acknowledged, in line with the editorial demands of Healthcare. The discussion of reliability coefficients, while technically correct, could be presented more concisely to meet the journal's standards of objectivity.
The results are presented in a coherent and consistent manner with the study's objectives, highlighting significant associations between psychosocial distress, social support, financial well-being, and gender. The interpretation of moderating effects is, in general, adequate and well-founded theoretically, especially in light of gender role theories. However, in some parts of the discussion there are extrapolations that could be better qualified, considering the impossibility of causal inference resulting from the cross-sectional design, as well as the punctual and imprecise use of the term "mediation" when the study exclusively tests moderation.
From a structural and editorial point of view, the manuscript follows the format required by the MDPI, including mandatory sections, ethical statement, conflicts of interest, funding, and author contributions. Nevertheless, there are significant weaknesses in the writing quality, with grammatical errors, repetitions, terminological inconsistencies, and excessively long passages, especially in the introduction and discussion. These problems do not compromise the scientific merit of the study, but they reduce the clarity and fluency of the text, which falls below the expected linguistic standard. A thorough linguistic revision and greater textual synthesis are necessary to make the manuscript more objective and editorially consistent.
The manuscript shows good adherence to the Healthcare journal guidelines (MDPI), both in terms of scope and scientific relevance. The theme of the financial well-being of informal caregivers of the elderly, articulated with psychosocial determinants, social support, and gender differences in rural contexts, is clearly aligned with the journal's focus on public health, social determinants of health, and vulnerable populations. The proposal is pertinent, current, and addresses recognized gaps in the literature, especially by incorporating gender moderation and focusing on rural caregivers, a group frequently underrepresented in national studies.
From a methodological point of view, the study demonstrates rigor appropriate to the cross-sectional design adopted. The sample is numerically robust and geographically diverse, the eligibility criteria are clear, the recruitment procedures are described transparently, and the statistical analyses are appropriate to the proposed objectives, with the correct use of regression models with interaction terms and control of relevant covariates. The use of validated instruments strengthens the credibility of the findings, and the limitations inherent in convenience sampling and self-reporting are explicitly acknowledged, in line with the editorial demands of Healthcare. The discussion of reliability coefficients, while technically correct, could be presented more concisely to meet the journal's standards of objectivity.
The results are presented in a coherent and consistent manner with the study's objectives, highlighting significant associations between psychosocial distress, social support, financial well-being, and gender. The interpretation of moderating effects is, in general, adequate and well-founded theoretically, especially in light of gender role theories. However, in some parts of the discussion there are extrapolations that could be better qualified, considering the impossibility of causal inference resulting from the cross-sectional design, as well as the punctual and imprecise use of the term "mediation" when the study exclusively tests moderation.
From a structural and editorial point of view, the manuscript follows the format required by the MDPI, including mandatory sections, ethical statement, conflicts of interest, funding, and author contributions. Nevertheless, there are significant weaknesses in the writing quality, with grammatical errors, repetitions, terminological inconsistencies, and excessively long passages, especially in the introduction and discussion. These problems do not compromise the scientific merit of the study, but they reduce the clarity and fluency of the text, which falls below the expected linguistic standard. A thorough linguistic revision and greater textual synthesis are necessary to make the manuscript more objective and editorially consistent.
Author Response
RESPONSE TO REVIEWERS COMMENT
Reviewer 4: comments
Comment 1: The manuscript shows good adherence to the Healthcare journal guidelines (MDPI), both in terms of scope and scientific relevance. The theme of the financial well-being of informal caregivers of the elderly, articulated with psychosocial determinants, social support, and gender differences in rural contexts, is clearly aligned with the journal's focus on public health, social determinants of health, and vulnerable populations. The proposal is pertinent, current, and addresses recognized gaps in the literature, especially by incorporating gender moderation and focusing on rural caregivers, a group frequently underrepresented in national studies. From a methodological point of view, the study demonstrates rigor appropriate to the cross-sectional design adopted.
Response 1: Thank you for your feedback and the time taken to review this paper. We appreciate.
Comment 2: The sample is numerically robust and geographically diverse, the eligibility criteria are clear, the recruitment procedures are described transparently, and the statistical analyses are appropriate to the proposed objectives, with the correct use of regression models with interaction terms and control of relevant covariates. The use of validated instruments strengthens the credibility of the findings, and the limitations inherent in convenience sampling and self-reporting are explicitly acknowledged, in line with the editorial demands of healthcare. The discussion of reliability coefficients, while technically correct, could be presented more concisely to meet the journal's standards of objectivity.
Response 2: Thank you for this constructive feedback. We have revised the manuscript to further streamline the discussion of reliability coefficients, presenting it more concisely and objectively in line with the journal’s editorial standards, while preserving the technical accuracy of the psychometric information.
Comment 3: The results are presented in a coherent and consistent manner with the study's objectives, highlighting significant associations between psychosocial distress, social support, financial well-being, and gender. The interpretation of moderating effects is, in general, adequate and well-founded theoretically, especially in light of gender role theories. However, in some parts of the discussion there are extrapolations that could be better qualified, considering the impossibility of causal inference resulting from the cross-sectional design, as well as the punctual and imprecise use of the term "mediation" when the study exclusively tests moderation.
Response 3: Thank you for this insightful comment. In response, we have revised the Discussion to more carefully qualify interpretations, explicitly acknowledging the cross-sectional design and avoiding any language that could imply causality. We have also corrected the terminology throughout the manuscript to consistently refer to moderation, rather than mediation, ensuring conceptual and methodological alignment with the analyses conducted. These revisions strengthen the precision and rigor of the discussion.
Comment 4: From a structural and editorial point of view, the manuscript follows the format required by the MDPI, including mandatory sections, ethical statement, conflicts of interest, funding, and author contributions. Nevertheless, there are significant weaknesses in the writing quality, with grammatical errors, repetitions, terminological inconsistencies, and excessively long passages, especially in the introduction and discussion. These problems do not compromise the scientific merit of the study, but they reduce the clarity and fluency of the text, which falls below the expected linguistic standard. A thorough linguistic revision and greater textual synthesis are necessary to make the manuscript more objective and editorially consistent.
Response 4: Thank you for this constructive feedback. In response, we have undertaken a thorough linguistic and stylistic revision of the manuscript. The Introduction and Discussion were carefully edited to reduce redundancy, improve clarity and flow, correct grammatical and terminological inconsistencies, and enhance conciseness. Long passages were synthesized to improve readability while preserving the scientific content. These revisions align the manuscript with MDPI’s editorial and linguistic standards and improve overall coherence and objectivity.